# Criminalized Intimacies between POWs and 'Unworthy War Wives' and Their Soldier-Husbands' Responses to Racial, Sexual Wartime Justice in Nazi Germany

Vandana Joshi

Department of History, Sri Venkateswara College, University of Delhi, New Delhi 110021, India; vjoshi@svc.ac.in or vandanajoshi2606@gmail.com

**Abstract:** The article places itself in the burgeoning literature on fraternization between POWs and local women during the twentieth century world wars. Though fraternization with the enemy was considered undesirable by all warring nations, incarcerating women for prolonged periods and suspending their citizenship for even longer was a hallmark of the Nazi system of (in)justice. War wives came in for harsher treatment as double traitors: to the nation and their soldier-husbands. The author has selected a few love triangles from a large cache of Gestapo reports; regional, local and 'Special Court' trials; and soldier-husbands' clemency appeals for a qualitative analysis. Her interrogation of the archival sources from a people's perspective goes into hitherto unexplored emotions, subjectivities and experiences of the affected people and deliniates how they appropriated, negotiated, rejected and defied the penal code in their own ways through a display of willful conduct. The interiority of their experience is juxtaposed with the discourse analysis of honor and shame surrounding criminalized intimacies to expose the gap between wartime discourse and social practice.

**Keywords:** criminalized intimacy with POWs; transnational wartime romance; fraternization; *verbotener Umgang*; emotions; race; gender; masculinity; National Socialist Justice; sexuality





## 1. Introduction

Fraternization with the enemy soldier has been a popular behavior in wartime societies and recent scholarship has explored it in various national contexts in the two world wars (Reiss 2018; Reiss and Feltman 2022; Todd 2011, 2017; Moore 2013; Feltman 2018). The latest edited volume of Reiss and Feltman (2022) has done a commendable job of initiating a much needed cross-cultural, transnational dialogue on fraternization as a universal behavior pattern in wartime. The contributors have variously called it 'collaboration of the heart', 'sexual treason', 'illicit sexuality', 'dishonorable conduct', 'sexual infidelity', 'treason combined with ideological transgression' and 'undesirable familiarity'. Across national boundaries and political dispensations, fraternization was considered unpatriotic, and undesirable. All warring regimes adopted a gendered approach to it in equating women's sexual purity with patriotism, and all of them engaged in propaganda campaigns to spread moral panic and fear of the racially alien enemy to protect inner national frontiers and prevent miscegenation. Women consorting with the enemy alien were socially ostracized, morally policed and culturally condemned as violators of the wartime gender codes of conduct. There is surely a prehistory to anti-fraternization in Germany. Todd, in her various works, in the context of WWI Germany calls it 'sexual treason', 'illicit sexuality' and 'sexual infidelity' and cites examples of fines and short prison sentences (Todd 2011, 2017, 2022). However, the legal implementation of sexual treason remained varied and inconsistent (Todd 2022, pp. 62–63). Neither the decision-making nor the punishments—fines and jail terms—were uniform throughout the empire. Other parts of the Western world too practiced a dual gender morality, developed a negative discourse around the war wife, launched moral campaigns and mobilized public opinion around it. Instances of gossip

mongering, fueling public outrage, publishing features and news items in newspapers, imposing fines, and less often, short prison terms were some of the ways in which fraternizing women were stigmatized, ostracized and punished for violating wartime gender codes. However, official and legal positions remained vague and uneven in these regimes too. The systematic way in which the Gestapo and the judiciary followed these cases until the bitter end remains unique to Nazi Germany. Its persecution of women offenders was embedded firmly in the legal system, and the way in which investigations were carried out gave the Gestapo functionaries not just immense power over the accused, but also perverse voyeuristic pleasure while extracting graphic details of their sexual encounters.

Given the history of fraternization during WWI (Todd 2011, 2017, 2022), the Nazi regime woke up early to the challenge by issuing the decree of 25 November 1939 amending the penal provision for the defense of German people. The article 4 of this decree stated, 'Whoever violates the provisions related to forbidden contact (*verbotener Umgang*) with POWs in a manner that severely damages the healthy racial instinct would be punished with imprisonment, or in serious cases, penitentiary'[1]. This was combined with the decree of 11 May 1940 barring all, except work-related, communication between POWs and Germans[2]. *Verbotener Umgang* may sound universally applicable to all Germans who sought contact with POWs. Its scope ranged from exchanging casual greetings and food items, delivering messages, seeking intimacy to aiding prisoners' escapes. Judicial verdicts evoked criteria such as undermining the fighting spirit of the German people, damaging the morale of the frontline soldier and displaying dishonorable and undignified conduct, repeatedly and shamelessly. However, not all exchanges and interactions resulted in long and harsh prison or penitentiary sentences. In principle, women related to soldiers, their wives and mothers, and those working in sensitive positions or the ammunition industry were punished with penitentiary. In practice, however, the large majority among these women were ordinary war wives with no previous history of crime.

Despite the much-hyped fear of espionage and escape, which invited longer penitentiary sentences and suspension of citizenship, it was the sexual and racial aspects of seeking intimacy that rolled out reams of paper from the Gestapo's typewriters and stacked up case files in courts. Criminalized intimacy, in practice a gendered crime, was the crux of *verbotener Umgang mit Kriegsgefangenen*. I choose to call this offense criminalized intimacies rather than *verbotener Umgang mit Kriegsgefangenen* because of its actual ramification. In the war years, the number of female criminals rose dramatically, and criminalized intimacies figured as the most preponderant crime among them. Johnson estimated that criminal cases against women more than tripled in the Gestapo records and nearly doubled with Special Courts, a large segment of which stemmed from their deviance from the racial, sexual and familial code of conduct, while Herbert called it a mass crime (Johnson 1999, p. 359; Herbert 1986, pp. 122–28). These evaluations are reinforced in studies dealing with *verbotener Umgang*. In addition to prolonged jail and penitentiary sentences, the verdicts led to the suspension of their citizenship rights and welfare benefits for a much longer period which meant the withdrawal of their separation allowance and other benefits. The actual process did not just involve the element of individual agency and political enthusiasm of particular functionaries for hunting down the deviants; the competing claims of the Gestapo, judiciary, RSHA (Reich Security Main Office), Reich's Chancellery and the SS made the actual process multi-layered and complicated.

## 2. Literature Survey

Previous works on *verbotener Umgang* can be broadly put into four categories: the discursive (Kundrus 1997, 2002; Schneider 2010), the empirical (Stephenson 1992; Boll 1991; Zühl 1992; Heusler 1995; Colmorgen and Godau-Schüttke 1995), the emotional (Usborne 2017) and the diplomatic (Scheck 2018, 2021). Not making a clear distinction between foreign workers and prisoners of war, the discursive strand argues that unlike the soldier, his wife attracted negative attention and publicity due to a consciously cultivated discourse aimed at controlling and disciplining her. The Nazi state's dual morality punished infidel

wives but let philandering soldiers get away. Appearing in the 1990s, it was a pioneering trend which Kundrus initiated. However, more recently, when it was revisited in Schneider's book length treatment, it lost its appeal. Schneider claims to mainstream *verbotener Umgang* by ordering the knowledge around it which is stated in the subtitle: *Diskurs um Sexualität, Moral, Wissen und Strafe* (A Discourse on Sexuality, Morality, Knowledge and Punishment). She takes us through a variety of sources such as brochures, advice manuals, press and propaganda literature, as well as other published sources to demonstrate the power and significance of discourse analysis. However, we do not see any real existing people, the targets of the penal sanctions who endured immense suffering, and who can be easily found in their thousands in German archives. This is not just a matter of chance but a conscious choice for her because individual agency or subjectivity is not her central concern. Using Foucault's theories and plunging deep into the text may be pleasurable sometimes, and useful at others, however, as Stephenson quips, 'examining a text in isolation from evidence that would either corroborate or contradict it—or even do neither—seems perverse' (Stephenson 2013).

The second category, namely local studies, offers an impressionistic view. Even they largely draw on the published Secret Service (*SD*) fortnightly reports, *Meldungen aus dem Reich* on public morale, media reports and official responses. With the exception of Boll and Colmorgen and Godau-Schüttke, who study archival dossiers of affected women, these accounts barely scratch the surface as they base their findings on filtered accounts of the persecuted in specific micro studies without processing their broader import. The third approach taken by Usborne explores emotions through judicial proceedings of the Special Court of Munich. Inspired by the works of Rosenwein (2006) and Reddy (2001), she assigns romantic love and erotic desire to adulterous German women and extends Frevert's (2011) discursive paradigm of honor and shame to their avenging soldier-husbands. Frevert, in her discursive treatment of honor and shame, argues that the Nazi regime purposefully appreciated the currency of honor both in racial and gendered terms. What was new in the way in which Nazism gendered emotions was that they structured and radicalized the already existing gendered emotions. They heeded women's emotions by evoking their reverence and support for the regime in a structured (controlled) way and then openly orchestrated them in the media. Men, on the other hand, held back emotions, controlled their passions and proved that they were capable of doing whatever had to be done without flinching. Frevert's focus here is not on how individuals coped with the challenges of navigating through the structures and prescribed codes of conduct, but on how institutions framed their emotional provisos and requests (Frevert 2011). Deploying this scheme in her essay, Usborne projects honor and shaming onto vengeful German husbands, thereby reproducing state-structured emotions mechanically as though husbands were mere instruments of the state. As evidence, she cites a vignette of a distraught husband's vengeful letter to his unfaithful wife, who as a professional soldier and party supporter, she presumes, would have drawn upon honor and shame to berate his unfaithful wife. He called his wife a 'shameless woman' and himself a 'broken man', implying that she had dishonored him with her 'shameful' and 'dissolute' behavior, which he avenged by threatening divorce proceedings and denying his wife any financial support (Usborne 2017, p. 465). She then goes on to explore two war wives' romantic love and erotic desire through the exchange of letters with their French lovers. Usborne's projection snatches all tender emotions from soldier-husbands, even though one can trace them in her hurried references to at least two other cases where husbands committed themselves to taking back their wives. Her binary ignores previously published essays on the same theme steering clear of evidence and perspective that put her framing in jeopardy (Joshi 2015a, 2015b). Theorizing husbands' emotions on the basis of a very narrow source base does not do justice to them when evidence from the dossiers more frequently points in another direction, as this essay will show. While Usborne gives long descriptions of romancing women, she does not accord their partners, lovers or husbands their fair share in 'her story'.

The last paradigm is represented by Scheck's monograph on Western POWs. He introduces a fresh perspective to the world of captivity and its connection with local people with a rich source base. He overturns the discursive world of the elite *Oflag* (officers' camps) prisoners who were fenced in and lived in an all-male environment. They were not obliged to work, and therefore, indulged in intellectual pursuits of creative writing, staging theater, making escape plans and sometimes also fooling around with guards and inmates to kill boredom. Ordinary prisoners, on the other hand, represented what he calls the 'alternate prison camp paradigm'. According to the Geneva Convention of 1929, their labor could be utilized by their captors which meant that working alongside Germans in the factories, on the farms and other sites requiring manual labor became the everyday reality of POWs in wartime Germany. Scheck's shift of focus from the much mythologized 'barbed wire university' to ordinary prisoners' everyday life who toiled alongside local people leads him to amass court martial records of POWs from French, Belgian, German, American, Swiss and British archives and for this task alone his work deserves praise. His search of ordinary prisoners in the foreign office records has made a significant contribution to an emerging trend among scholars who use diplomatic archives not for writing traditional history of high-politics but for producing narratives of neglected and marginalized groups (Furuya 1995; Krebs 2015; König 2019; Lorke 2019, 2020). Scheck's later descriptive treatment of Special Court cases, however, leaves ample scope for research in new directions and this paper aims at doing just that.

### 3. Space, Context and Research Problem

World War II completely transformed Germany's human landscape. By the summer of 1944, 13 million German men were drafted and 7.6 million aliens, including about 2 million captives, joined the workforce (Herbert 2000, p. 199). They came from diverse nationalities and linguistic backgrounds and were housed in various types of camps: guarded *Stalags*, labor commandos often inside the workplace in abandoned buildings and in rural areas often scattered and in close proximity to small and middle-sized farmsteads. Yet, in the post-war memory scape they were almost invisibilized in memory projects. Thus, when Ulrich Herbert conducted oral history interviews in the 1980s in the Ruhr, a predominantly working-class area, which was overwhelmed with POWs and forced workers in WWII, he noted that respondents only talked about the aliens when asked, not voluntarily (Herbert 1983). Aliens were not an obvious part of their memory scape when their own survival was at stake. Given the spread of criminalized intimacy in the working-class milieu, it was womenfolk who saw (or looked away) and experienced (but suppressed) the presence of aliens in their midst, and definitely had more insights to offer. Yet, Herbert himself chose only three female and seven male respondents for his memory project, which indicates a methodological oversight (in which a worker is essentially conceived as a man) is not uncommon to social historians. It is not just oral history respondents and interviewers, rather, even feminists, who rewrite military history, have been great naturalizers of methodological nationalism in presenting soldiering and the home front as nation-specific case studies (Herzog 2009; Higonnet et al. 1987). They split the universe of war along the fault line of gender but retain the nation in their research method as an unproblematized analytical unit. The cross-cultural, transnational context of working people within the boundaries of the nation-states during war years is thus blended out. This paper locates ordinary prisoners in the feminized home front as workers during the war years where they labored alongside German women on the factory floors, farms, railway stations and other sites such as gardens, forests and so on.

German women and POWs thus practically lived in a transnational working environment, breaking the mold of what Martins calls methodological nationalism (Martins 1974, p. 276). Although framing people's history as transnational history has since been a stock in trade for migration studies and postmodern discourse, it is fairly recent to war studies and a welcome turn (Rass 2016; Walleczek-Fritz 2016; Joshi 2015c; Reiss and Feltman 2022). Martins' transnationalism is a useful method of conceptualizing working people—in this

case, German women and POWs, from diverse cultural and national backgrounds—and their space and time. It opens up the opportunity to study the emergence of new patterns of sociability, albeit forbidden, related to work, leisure, love and war that became intrinsically intertwined. This transnational perspective allows us to see multitudes of ordinary people jumping out of their racially, nationally and sexually divided silos and moving about in a floating zone: the ultimate liminal space (Reiss and Feltman 2022, p. 8) of *verbotener Umgang* where they hid themselves, romanced, exchanged gifts, romanced, had sex and quickly disappeared.

The paper further seeks to critique the discourses of honor, shame and romantic love within which criminalized intimacies have been conceptualized until now to write the history of emotions. It follows Mandler's (2004) approach, methods of micro history (Ghobrial 2019; Ginzburg et al. 1993) and the notion of *Eigen-Sinn* (Lüdtke 1993, 1995) to raise a different set of questions and seek different answers. While dwelling on the problems with cultural history, Mandler fears that the discursive exercise could distract us from our responsibility to evaluate not only the meaning of the text but also its relation to other texts, its significance in the wider discursive field, its relation to its 'throw' and its dissemination and influence, that is, the conditions not only of its production but also its distribution and reception (Mandler 2004, pp. 96–97). In this article my engagement with the 'throw' is combined with methods of micro history, to consciously prioritize the emotions, subjectivities and experiences of marginalized and persecuted people, in our case, people who came in the ambit of criminalized intimacies, over those of the high and mighty who structured emotions. I borrow the term *Eigen-Sinn* from Lüdtke, the exponent of the history of everyday life, which denotes willfulness, spontaneous self-will, a kind of self-affirmation and an act of (re)appropriating alienated social relations on and off the shop floor by self-assertive prankishness, thus demarcating a space of one's own to understand the behavior of these people. Lüdtke proposes that there is a disjunction between formalized politics and the prankish, stylized, misanthropic distancing from all constraints or incentives present in the everyday politics of *Eigen-Sinn*. Though the term has pejorative overtones, referring to 'obstreperous, obstinate' behavior, usually associated with children, the 'discompounding' of writing it as Eigen-Sinn stresses its root signification of 'one's own sense, own meaning' (Lindenberger 2015). I find it useful to understand the behavior of people who ran against the prescriptions and proscriptions of criminalized intimacy despite being fully aware of their 'wrong doing' during their interrogations.

Their *eigen-sinnig* defiance to the racial-sexual decrees of *verbotener Umgang* created a different range of experiences, subjectivities and emotions that set them apart from the law abiding 'normal' people. They sought refuge in unconventional deeds and spaces to realize their desired goal. They played the forbidden game of desire which created its own semiotics of rebellious romance, hibernating at times and traversing through a hostile terrain at others, navigating with the help of codes and practices that they devised to sustain their romance.

This approach raises a set of questions: How were these state-structured emotions, namely honor and shame, received by those who became their targets? How did they respond to the prosecutors? Was the criminalized intimacy really undermining the front soldiers' will to fight the enemy, as court verdicts claimed? Or was it the persecution of their wives that caused them horror, pain and agony? Can we reframe suffering (Hansen 2011), or demoralization in this context, from the perspective of the actual sufferers? In other words, as soon as we shift the focus to the 'throw' of criminalized intimacies, an entire new world of possibilities, narratives and archival sources opens up. They offer a range of lived and embodied emotions that do not feed into the discursive regime of honor and shame. An evaluation of the 'throw' through the hitherto unexplored archives on criminalized intimacy promises to take us away from the propaganda and advice manuals as well as Gestapo reports and court verdicts, to the homes of war couples and the sites of criminalized intimacies.

For this paper, I have selected a few cases for a qualitative analysis of long-standing love triangles from my diverse collection of trial records. This selection is based on the richness of the dossiers so as to gain access to the subjectivity, agency and emotions of ordinary people caught in extraordinary circumstances, and especially those of soldier-husbands who successfully rescued their wives from the clutches of their persecutors. In what follows, it will be demonstrated how individual actors who were caught in the net of criminalized intimacies first defied the sanctions and then appropriated, negotiated or rejected the normative sexual-racial codes, exposing the cracks in the hegemonic discourses of the emasculated POW, subservient war wife and the hyper-masculinized German soldier. Before I do that, an evaluation of the archival sources seems to be in order.

## 4. Archival Holdings

In the initial war years, women were subjected to open humiliation by party workers who pilloried them with shaven heads and hung humiliating placards around their necks calling them race traitors, yet their numbers kept increasing (Kasberger 1995; Gellately 1990, 2001; Muggenthaler 2010). Prudent policy makers argued that random violence by party zealots invited public displeasure and bad international publicity rather than compliance. Jurists advocated the restoration of order by initiating criminal proceedings. Gestapo investigations and courts trials became the order of the day until the fall of the Third Reich. This systematic and focused targeting of women for their 'sexual-racial defiance' produced thousands of case files on private individuals which still survive in several German archives despite instructions from above to destroy all incriminating evidence against the regime as surrender became imminent.

Roland Freisler, the State Secretary in the early years of the Reich who went on to become the dreaded President of the *Volksgerichtshof* (People's Court) in 1942, exhorted Special Court judges to act like 'panther troops': speedy, forceful and efficient in the dispensation of justice to the internal enemy of the home front in the war years (Wrobel and Maul-Becker 1994, p. 16). *Verbotener Umgang* was tried among other wartime offenses such as deceit, plunder, theft, forbidden slaughtering of animals, crime against property, listening to foreign broadcasts and slanderous gossip for undermining the fighting spirit of the German people. Available studies on *verbotener Umgang* (Wrobel and Maul-Becker 1994; Weckbecker 1998; Keldungs 1998; Colmorgen and Godau-Schüttke 1995; Boll 1991; Usborne 2017; Scheck 2018, 2021) based their finding on Special Court trials, as did I in my initial stage of research which evaluated *Sondergericht* (Special Court) holdings in the *Niedersächsisches Landesarchiv, Hannover*. It contained 146 dossiers, which was the largest category of wartime offense after plunder and theft. The Hanover case files represent a mix of rural and urban population and a diverse denominational base, making it a suitable area to study average behavior patterns. However, to broaden my source base, when I visited the *Landesarchiv Berlin* (Regional Archives, Berlin), I discovered that it contained approximately 886 dossiers of the *Landesgericht* (Regional Court) and 176 of the *Amtsgericht* (Local Court) on *verbotener Umgang*. Several dossiers dealt with groups of accused women, while some additional ones were created for clemency appeals. This indicates that even though the Special Courts tried a majority of these cases, *verbotener Umgang* became grist to the mill of all courts, at least in the case of Berlin, which makes the spread of reported instances much wider than believed. It will require further research to establish if other larger cities followed this pattern. Having said this, it is important to keep in mind that several cases might never have been reported, especially in rural areas where isolated farmsteads could not be closely supervised, and consensual sex could have been kept under wraps.

All case files were prepared by the Gestapo who worked in close collaboration with the prosecutors. The Gestapo exercised their own arbitrary power on the accused through arrests, searches, evidence collection, protective custody and interrogations. They dismissed several cases after investigations. Apart from checking the political, racial and confessional background of the accused as a routine, their investigations could involve interrogating

neighbors and colleagues, and seeking the expert opinion of doctors, 'racial biology experts' and psychiatrists to create a psycho-sexual profile of the accused. In cases involving abortions and deliveries, they reached out to midwives, doctors, social workers and the youth office in the process of the investigation and preparation of the case files for trials. Because the network laid out was very elaborate, the surviving records offer relatively much richer material to study fraternization in comparison with any other national context.

Despite being generated by the oppressor, these trial cases are a valuable source for historians. They contain traces of an entire range of emotions experienced by defendants, their kin and other involved parties that offer us invaluable insights for writing the history of emotions. While we can directly read the mind of the oppressors, their disbelief, suspicion or exasperation in the textual evidence such as investigation reports, witness testimonies, Gestapo's case summaries and court verdicts, it is far more difficult to read the victims' deeper feelings as they tried to conceal them at first and owned up to them under duress in subsequent rounds of interrogation. The narratives penned by the Gestapo and judiciary reek of condemnation and disrespect for emotions such as sympathy, compassion, curiosity, romance or adventure that women displayed towards the alien enemy and noted them as evidence for their dishonorable conduct at best or psycho-sexual disorder at worst.

However, there are other pieces of evidence that help us reassemble the semiotics of romantic and conjugal love, desire, despair, fear, longing and loss. The especially thick dossiers contain objects that were seized during arrests and searches, such as exchanged gifts, pictures, memorabilia and love letters or their description. These textual and material objects which were not meant to fall into the prosecutors' hands are embedded with instrumental and amorous value that reveals the emotional world of the victims. They encapsulated their phantasies and desires. Although a deeper exploration of the seized material is beyond the scope of this paper, I will point towards its import while presenting my cases and generalization.

More interestingly, the dossiers contained letters and indirect communications through lawyers and well-wishers that defendants' soldier-husbands sent, mostly in the form of clemency appeals. They are the focus of the present study. These appeals are rare archival deposits that have previously not been used by any historian. They display a range of conflicting emotions of soldier-husbands towards their partners, emotions that seldom surface in ego documents such as letters from the front (*Feldpostbriefe)*, memoirs and autobiographies. Considering the strong taboo that sleeping with the enemy carried, no soldier would have voluntarily liked to write about it in an ego document during or after the war. I understand all these fragments of evidence generated by the affected people as popular scripts against the textual sources that scholars use for discourse analysis: advice manuals, prohibitive notices, propaganda literature and news items that constantly reminded them the 'enemy remains enemy'. Through unpacking the layers of these clemency appeals, as well as reading the clues from the seized objects, this paper, for the first time, attempts a wartime history of emotions from the perspective of war wives and soldiers, both captives and soldier-husbands, who endured immense suffering at the hands of an unjust system. While doing so, I indulge in a dialogue with scholars who are working on emotions, subjectivities, individual agency, gender and masculinity in and outside the context of wartime Germany (Langhamer et al. 2020; Roper 2005; Noakes 2020; Stargardt 2010; Hansen 2011; Jarausch 2011); those who are engaged in writing the history of interracial relations during Nazi Germany (Krebs 2015; König 2019; Lorke 2020; Furuya 1995) and those who are working on local women and POWs in a transnational context of wars (Reiss 2018; Reiss and Feltman 2022; Todd 2011, 2017; Moore 2013; Feltman 2018; Scheck 2021).

## 5. Case Studies of Love Triangles

Frieda K.'s dossier of the Special Court proceedings from the Hanover archives is extraordinarily rich in detail and shows the working of the prerogative state (Fraenkel 1941) with its full potential as well as its limit. It also unravels the story of a miraculous

escape from the penitentiary. Frieda was strongly suspected of having borne the child of Gabriel C., a French POW. The case started with a denunciation submitted to Gabriel's prison camp commander at Steinhude on 9 April 1942[3]. Gabriel C. was accused of having written a letter to Frieda that proved his paternity. His letter was forwarded to Wunstorf for further action. The report stated that the suspect gave the letter to his co-worker Elfriede K. to deliver to Frieda, but Elfriede K. read it and handed it to Hilda P., who gave it to the camp commander. All of them worked in a wholesale food store in Fallingbostel. The camp commander also noted that Gabriel C. denied having written the letter.

Hilda P. was interrogated by the Gestapo on 24 April 1942. She told them that Frieda and Gabriel C. were her colleagues. There were rumors that Frieda had a relationship with one of the French prisoners but there was no concrete evidence. Elfriede showed her the letter saying she should keep it with her, should the matter of their relationship come up again. Others advised her to hand it over to the prison camp post. When asked whether Frieda delivered a baby, she said yes, but could not say with certainty who the father might be. The second witness Elfriede said that anyone could tell from their behavior that there was a relationship between the two, but they had no proof. In early April, Gabriel C. came to her and slipped a letter into her pocket. It was addressed to Frieda. Since she had the same nickname, she thought it was for her and read it. The next day, she showed it to Hilda who handed it over to the camp post. As Frieda's neighbor, she knew that Frieda had delivered a baby in January 1942. It was clear to her from Gabriel C.'s letter that he was the father. She also knew that Frieda was in love with a German soldier earlier but could not tell when the relationship ended. Frieda herself told her that he had not been on leave for more than one year before his recent visit, so she assumed that he could not have been the father. She also thought that the child greatly resembled Gabriel C.

Frieda, a twenty-two-year-old mother and the suspect, was taken into Gestapo custody for interrogation. She completely denied having intimate relations with Gabriel C. Gabriel and his paternity. She claimed that the baby was from an unknown soldier whom she got to know on 1 June 1941 in Steinhude. When reminded that she was often warned by the colleagues, she said it was just because she gave him bread. The Gestapo concluded that she was a shameless, totally untrustworthy liar, with a bad upbringing.

The accused Gabriel C., a 24-year-old French prisoner, first denied all interaction with Frieda but, during his court martial, confessed to his relationship and also owned up to the letter. Soon enough, he prepared another statement and sent it to the military court pleading not to punish Frieda and instead deploy him on the Eastern front. He also claimed that he intended to marry her and had applied for naturalization. It was not just because he loved her but also because he sympathized with Germany. He even informed his parents about his intentions to marry a German girl who had borne his child and they consented. He gave the date and place of their intercourse as late June 1941, the child's date of birth, and stated that he knew that she had had a fall shortly before her delivery and that it was a premature baby of 7 months. He dismissed Frieda's claims that someone else was the father. He was confident that she wanted to marry him. Based on his confession, he was sentenced to a prison term of 2 and a half years.

Meanwhile, Frieda sat in Gestapo custody and was to appear at the main hearing at the Special Court on 6 October 1942. She later named the anonymous soldier as Willi L. In the meantime, the prosecution conducted a blood test on the mother, child and Gabriel C. The test results did not rule out Gabriel C.'s paternity. Then, the story took a dramatic turn. A thirty-one-year-old Willi L., stationed in France, approached the prosecutors voluntarily on 17 December 1942 and declared that he was the father of the child and Frieda, his fiancé, was his childhood love. They had had sexual relations for the past 3 years including the three-month training period before he left. The last time was on 1 June 1941 which he remembered clearly as he had to leave on 2 June. He absolutely ruled out the possibility of another man in her life, let alone a POW. They could not get married earlier as their papers were not ready. He pleaded for her immediate acquittal as he wanted to marry before his leave ended on 1 January 1943. His request was granted immediately, and they

were married soon after. A blood sample was collected from Willi and the report stated that his paternity, just like that of Gabriel C., could not be ruled out. The way the paternity tests were reported in the dossier showed that the method worked on the basis of exclusion rather than precision. The child's blood report contained OMN written together. The blood report of Gabriel C. had O (a gap) and MN next to each other. Frieda's test report contained A1 and M, while Willi's had A1 and MN. The conclusion derived from this was that the child's factor M came from his mother and because Gabriel C.'s blood type had N next to M, the child received the N factor from his father's side. His paternity could only be ruled out if he either had the blood group AB or only the M factor. Since Willi's sample contained neither the AB blood group nor an exclusive M factor, his paternity could not be ruled out either.

In view of Frieda's denials of Gabriel's paternity, the prosecution followed up Gabriel's case with more rigor. He was shifted from one camp to another before landing in Graudenz. After Frieda's acquittal he was interrogated again. He kept insisting on his love for her and his marriage plans with her. During his last interrogation on 9 April 1943, he carried a picture of Frieda and broke down. He showed his willingness to confess his involvement under oath, if required. In July 1943, an *SD* (Security Service) man was sent to France to visit his parents who confirmed his intentions of naturalization and marriage to Frieda. Their obsession with Frieda did not end either. They got in touch with her midwife and doctor who confirmed that it was a premature baby, making it difficult to ascertain the time of conception. The doctor put the date bracket from mid-June to mid-July rather than early June, again pointing the needle of suspicion to Gabriel. They then contacted the youth office, interrogated more colleagues from the workplace and also contacted the Kaiser-Wilhelm Institute for Anthropology in Berlin for the child's racial-biological examination. The institute wrote back in July 1943 saying that any assessment was not possible until the child was 2 years old. The criminal case had to be dropped against Frieda on 14 June 1944 after a wild goose chase on the grounds that Willi had already taken her as his wife and claimed the child's paternity.

What is extraordinary about this long and winding story is the prosecution's simultaneous obsession with, and frustration at, the incorrigible attitude of the involved offenders, especially Frieda, who refused to admit her guilt. In most other cases, the Gestapo, through its notorious methods of intimidation, a confrontation with the lover or other witnesses and the presentation of incriminating evidence broke the resistance of the accused and forced a confession from them which became the basis for a trial. When these methods failed to crack the case, they resorted to other means, such as taking blood samples of the suspects, sending an SD officer to Gabriel's home and requisitioning a racial-biological test on the infant, yet they could not derive the pleasure of having solved the case.

What is quite common about Frieda, Willi and Gabriel is that they were ordinary working class people whose conduct railed against the state's prescriptions and proscriptions. The Nazi regime practiced racism and sexism not only on their womenfolk and enemy aliens but also on their collaborators, such as the Italians and the Japanese who had relations with them and wished to marry (Furuya 1995; Krebs 2015; König 2019). On the other hand, it encouraged the birth of racially desirable 'Nordic' children from Norwegian women in and outside of marriage with German soldiers. The numbers of these desirable children born in Norwegian *Lebensborn* homes rivaled those of their German counterparts within four years (Nilsen 2019, p. 187). So, while the Nazi regime was constantly finding racially worthy people who could be Germanized in the occupied territories, it was simultaneously building racial boundaries for the people of alien blood.

Frieda's case of dramatic acquittal, despite all odds, on the insistence of her soldier-lover is as surprising as it is indicative of how such appeals may have shaken the establishment. Soon after her release, a memo was sent on 14 January 1943 by the Ministry of Justice to all public prosecutors with clear instructions on how to deal with cases of *verbotener Umgang*. This five-page-long confidential document advised the judiciary to assess the cases on the basis of the intensity of the woman's dishonorable conduct and the

depth of their bonding with the POW. Married women had to be sent to a penitentiary if they were soldiers' wives, mothers or kin; if they initiated the relation; or sustained it over a long period. The same applied to women working in armament factories or sensitive positions, and if they were privy to prisoners' hostile stance towards Germany. The latter cases were, however, rare. The last paragraph pointed out that if a soldier-husband, despite his wife's sexual relations with a prisoner, wanted to forgive her and continue the marriage, his clemency appeal be accepted and his wife's sentence be suspended. However, his wife had to be made aware of the reason of her release and the soldier might be given a convalescent leave[4]. Willi's action may take the reader by surprise as it goes against the image of a German soldier as hyper-masculinized, resolute, brutal and emotionally in control of himself, but we will meet more such men in the following pages and make sense of their responses.

A 32-year-old Hilde B. was sentenced to 8 months imprisonment on 12 September 1944 by the Local Court of Berlin[5]. She was arrested on 15th August 1944 for her involvement with a Belgian POW, Gascon D. Hilde's house was bombed out in January 1944. She was relocated with her minor children to the countryside and billeted with a peasant on whose farm Gascon worked. Her everyday interaction with Gascon developed into an affair. During her interrogation, she confessed to kissing him and having sex with him twice. The jury indicted her harshly as a war wife and mother of three minor children for cheating on her soldier-husband and disregarding racial proscriptions, willfully and repeatedly. Within 3 weeks, her soldier-husband Otto B. submitted a clemency appeal personally to the public prosecutor citing the following reasons:

> My marriage to Hilde completes nine years now. We have three children from the marriage in the ages of 3, 4 and 7. I myself have been a soldier for three years and am posted on the Italian front. My children have practically become orphans due to my absence and my wife's incarceration. I moved my children with my siblings, but this option can only be temporary as they have to go to work and have other members in the house. In favor of my wife I must say that my children are very attached to her. The same is true of my wife. They suffer a lot in her absence and have to pay a price for her follies unwittingly and inadvertently. So far as their upbringing is concerned, they are faced with major disadvantages. It is very difficult for me to send them to an orphanage as it will cause me extreme despair. My wife and children were my life's fortune till now. Destiny has played a cruel joke on me and has rocked my married life. I hope to bring the situation back to normal. I hope to bring back the mother to my children and my wife to me.

> As far as her wrong doing is concerned, I am the best person to testify on her hitherto conduct in marriage. She has always been a loving companion to me, a good and capable housewife, and most of all, a caring mother. She is by nature a generous soul, full of compassion and sympathy for fellow humans. She lost her brother at the Eastern front last year. Our home was consumed by flames in a terror attack last year. Since then, she has been wandering from one place to another restlessly in search of a shelter. Her relocation with the children to the inhospitable circumstances in East Prussia has claimed her poise and self-confidence. It brought her to a point of becoming vulnerable, dependent and in need of support. She bonded with a man who pretended to sympathize with her suffering to come close to her and used her for his evil design. In her boundless generosity and sheer foolishness, she unfortunately succumbed to her own sentimentalism and got trapped into an unscrupulous web.

> Her wrongdoing has affected me most deeply. She was my greatest pride. Unfortunately, a big disappointment has befallen me now, which is going to burden me throughout my life. I am pleading for clemency today without being pushed by her and despite the innermost suffering she has caused my soul. I do so because

of her innate good-heartedness and an assurance from her that she has found her true self once again. I too am convinced of it. Who would otherwise want to degrade and humiliate himself by seeking to revoke a penal sentence against his unworthy wife? Not underestimating the inner worth of my wife, I have once again reached out to her to give her an opportunity to make amends.

If for nothing else, I would like to get the mother back to my children and my wife back to humane surroundings. I hope she would be in a position to assuage my deep feelings of hurt by her gratitude towards me. Given her absolute lack of exposure to people, her naivety and kindness, if I allow her to fall, she will be doomed. She is so unhappy with herself that she will not be able to forgive herself for the loss of her children. She is extremely depressed about her incarceration, and so far as I know her, will take with her a bitter lesson for a lifetime. The time that she has already spent in jail is enough to warn her from repeating the wrongful act ever again. I am prepared to forgive her folly and would be grateful to get her a clemency through my appeal and to be able to bring her to my children soon.

I will be going back to the front on 8 October 1944. It would be a great consolation for me to know that my children are once again in their mother's care. By granting clemency you would also be strengthening me in my joyful deployment for the great German Reich.

Hail Hitler

Otto B.

Though the jail authorities did not favor an early release, her sentence was cut short by 4 months due to Otto's intervention. The irony of these soldier-husbands was that the war resulted in prolonged delays in their homecoming while the empire 'came home' in the form of alien captives. Conventionally, it was men who went out to conquer, enslave and indulge in sexual adventures. WWII, much like WWI, created a home front which gave women prolonged exposure to aliens in a close working environment and generated new sexual dynamics of independent, sexually discontented and adventurous wives and absentee soldier-husbands (Herzog 2002, 2005, 2009). Contrary to the state's assertion that such socialization had to be criminalized in order to prevent espionage, flight of prisoners, sexual promiscuity in the female population and race pollution, the large majority of captives neither attempted escape nor espionage. Both the captors and captives were more inclined to making their work environment humane, adding humor, ease, adventure, sex and romance to their war-weary existence.

The next case illustrates the conflicting emotions of a solder-husband caught in similar circumstances:

A 27-year-old woman named Maria P.'s soldier-husband Paul P. reported a French POW Rudolf T. to the criminal police[6]. Maria's testimony to the local court (*Amtsgericht, Berlin*), revealed a love relationship between Rudolf and her that developed in the course of her evacuation from a bombarded house in late 1943. He was hanging around the barracks when she asked him for help. Maria was grateful for his help in relocating her, bringing her a Christmas tree and giving her a security lock for the entrance door. When she asked what she owed him for all that, he said, 'a kiss', and she obliged. He declared his love for her in a letter and the affair led to making love at her conjugal home. Her home was bombarded again, and he volunteered to help again. When her soldier-husband came home on leave in May 1944, he discovered the love letters and denounced him to the camp commander. The matter was forwarded to the Gestapo.

During her interrogation, Maria not only confessed her love for him but also said that she had a better understanding with him than her husband and that they had marriage plans. She was jailed for six months in June 1944. Soon thereafter, her husband made a handwritten clemency appeal. He wrote that he would go to the front as half a man because his wife had been taken away from him and he feared his minor children would land in

alien hands. Their house was bombed out and his children were in no position to handle the required paperwork. How could he perform his duty peacefully at the front in these circumstances? He did not want to deliver his wife to prison authorities at any cost as the entire family's existence depended upon her release. In July, he sent another letter which led to her release on probation.

As early as October 1942, the *SD* noted that prisoners of war were earning a good reputation in their workplace and surroundings and that this tolerance was being misused to get close to sympathetic womenfolk to enter into sexual relations with them. This was not just because of working together but because women no longer maintained the required racial-political distance from them once they won their trust. A majority of these women came from socially and 'racially good' families in the countryside. The *SD* observer felt that not enough was being done in the media and radio broadcasts to address the danger of fraternizing with the enemy and suggested enhancing racial awareness through films, placards and word of mouth publicity by party workers. In December 1942, again, he showed concern about the laxity with Western prisoners, and a lack of racial awareness and Catholic brotherhood[7]. The *SD* report observed with regret that Germans considered the French as diligent co-workers and co-religionists in Catholic majority areas. They even claimed that German women agreed to sleep with camp guards if they allowed French POWs more freedom to move in and out of camps. Romantic relations between them and German women multiplied in the later war years. They were depicted as the most sexually active and desirable colleagues who initiated sexual contact with German women at work[8].

It was regularly noted in the reports that French POWs were the largest group among them, enjoyed relatively more freedom and also initiated contact. Their number in Graudenz, a notorious punishment camp where Gabriel C. was sent, was also the largest (Scheck 2021).

The Polish and Russian Decrees of March 1940 and February 1942, respectively, under Himmler's instructions, ordered the public execution of Polish and Russian POWs in front of their camps while women were sent to concentration camps (Muggenthaler 2010). These incidents were widely publicized in the local press and Nazi newspapers to instill fear in the minds of people. These orders were revoked, and the law was allowed to take its course from early 1943. Nonetheless, sentences remained much harsher for the Eastern prisoners. Unlike the Western prisoner who had protecting powers, they came under the Supreme Command of the German Army, and when found guilty they were handed over to the Gestapo who sent them to concentration camps for prolonged periods. This terror tactic substantially reduced their numbers, as my samples from Hanover show. Out of 146 total cases, 118 involved Western, 26 Eastern and 2 Jewish POWs. Most of these cases of Eastern POWs came from the countryside where they were not closely supervised and could secretly sustain their relations in a barn, farm or the lone lover's home until they were exposed due to accentuating circumstances such as a pregnancy, attempted abortion or the accusation of rape by the pregnant women, as in the following case:

A 24-year-old farmer's daughter, mother of a minor child and soldier's wife, Anna M., was accused of having intimate relations with a Serbian captive Stanislav W. who had been working on their farm since July 1941. Anna was the only family member working on the farm with him as her parents were old and sick. Their constant interaction at work brought them closer. He showed an interest in her, and by early 1942, their friendship resulted in mutual kissing and hugging. By August, they indulged in sex repeatedly in the barn. In December 1942, she had a miscarriage. She contested attempting an abortion on her part. She also confessed that she did not have the power to fight the captive's repeated demands. The Special Court remained unconvinced and sent her to a penitentiary for 18 months in a hearing on 19 February 1943[9].

In May, her husband sergeant Kurt B. expressed his opinion on his wife's conduct to the jury as follows:

> I am told that my wife had relations with a Serbian prisoner for which she has received a term of 18 months in penitentiary. After much thought, I have decided to forgive her. I cannot comprehend her misdemeanor. From the proceedings I

have gathered that she herself could not understand how she could do it. The Serb must have exercised some incomprehensible power over her that weakened her will to resist. I fail to make sense of how my wife could have behaved in this crass manner. She is weak-willed and can be influenced easily. My opinion has been reaffirmed by what I recently experienced during my last leave. I realized that my wife had made all possible efforts to create obstructions in the Serb's way of approaching her. A village woman told me that my wife had requested her to stay on till the Serb returned to his camp after work. I definitely want to maintain my marriage especially in view of our 3-year-old son. My defense lawyer has submitted a clemency appeal on my behalf. I have been a soldier since 3 February 1941 and am currently posted in Leningrad. To make matters worse, my family lives with my parents-in-law on their farm in Steinau, and my wife's presence is required there very urgently. My Father-in-law is sick and my mother-in-law is very old. She is tied up with my son, so the whole farm is left to Eastern prisoners.

Hail Hitler

Kurt B.

Kurt B.'s first appeal did not yield any result. Then, he appealed directly to the Reich's Chancellery on 23 December 1943 and was able to get her released on parole.

In another case in 1943 involving a Serbian POW, a German war wife, carrying his child, accused the Serb of rape, but he claimed that their relationship was consensual. The jury did not trust her; nonetheless, she was acquitted because her soldier-husband, posted at the Eastern front, refused to believe that his loving and conscientious wife could ever behave in this manner and was only willing to believe her version of the story[10].

While the wives may have been rescued by their soldier-husbands or their lovers, the prisoners, if they were from the South and the East, in particular, were left to the whims of the Gestapo. I was able to trace one such case of Martina B's Serbian lover in the ITS archives. He was liberated by the US army from the Buchenwald concentration camp, where he was incarcerated for four years. He was a single 24-year-old prisoner who worked on Martina's father's farm. Martina, a mother of three minors and a twenty-eight-year-old widow, developed a longstanding relationship with him[11]. When he was transferred, Martina gave him 24 self-addressed envelopes, one of which was intercepted by the Gestapo. During the interrogation, he accepted the relationship and argued that he was not a Serb but a Croatian, and that they were fighting on the German side. He saw nothing wrong in the relationship. He was serious about her, had promised her marriage and she trusted him. The Gestapo were initially not quite sure how to handle this blue-eyed boy (the ITS archives keep data on physical features among other things, unlike those of the judiciary) and spared Martina the initial arrest. However, finally she had to give her children into various homes, and he landed in Buchenwald. When he was liberated, he had three missing teeth[12]. I could not gain access to his life after the liberation due to the Data Protection Act. Another case of a Belgian who was involved in a serious romantic relation could not be accessed at the ITS on similar grounds.[13]

Even the military courts were not immune to husbands' clemency appeals. An Air Force News Service employee, Käte H., became involved with a Polish captive, who worked on her uncle's neighbor's farm. She met him in 1940 during a visit, maintained contact with him through letters and traveled back to meet him again in June 1941. After returning home, she sent him a letter in July 1941, which led to the discovery of their relations. She confessed that she was fond of him and had kissed him a few times. Her court martial resulted in 6 months imprisonment. Her husband's appeal got her out of the prison within three months on parole[14].

In an extreme case, Eric W. first denounced his wife Martha to the Gestapo on 23 July 1943 for sleeping with a French prisoner in their marital house. She was arrested immediately and sentenced to 2 years penitentiary on 19th October 1943. When she was

transferred from custody to the penitentiary, she was already a widow. Eric committed suicide in August 1943. Martha appealed for clemency on 29th August saying that he was extremely sorry for having denounced her and took his life. She needed to attend his last rites[15]. The court gave her leave to do so.

These representative cases show how different the subjective perceptions and emotional experiences of these actors were from the state-structured emotions of honor and shame. The testimonies of lovers exhibited the awareness of the transgression, yet they sustained the relations as far as they could. War wives saw in these prisoner-lovers young, brave soldiers, husbands and fathers, quite like their own, but who were available and willing to offer them their labor and love. Their acts, thoughts and spirits did not correspond with the image of the enemy alien propagated by the regime. Statements that war wives made about them during their interrogations bear testimony to that 'He is also a papa', said a mother to her little child when he exclaimed papa! on seeing a POWs while going for a stroll. They soon turned into lovers 'He liked me and I did not resist his moves. I had not had sex in a while', said another 'When you are beside me, I feel totally different from when I am with X (her Husband)' wrote yet another war wife in a letter to her lover. The physical separation from soldier-husbands whose leave became less and less frequent, and the help rendered by the POWs on a daily basis provided the context in which alien co-workers became desirable romantic lovers. War wives responded reciprocally to their requests, and even initiated these relationships. They skillfully used their resources such as their marital homes to host POWs, or take them to secure surroundings, bring them food and clothing, mend and wash their clothes and assure mobility for them by purchasing train tickets or giving them their husbands' clothes to camouflage their identity. The seductive charm of alien captives and their eagerness to help brought hope, adventure and excitement in the lives of overburdened war wives, whose workdays were filled with anxiety, fear and gloom. These were stories of the mutual bonds they developed to fight the loneliness and drudgery of a protracted war.

For the POWs' experience, I come back to Gabriel C.'s story as being representative of several others in spirit and acts. Gabriel C. was repeatedly threatened by his camp guard with harsh sanctions if he did not mend his ways, but he did not care. Not many among those who indulged in romantic relations with war wives were cheeky enough to boldly state their intent and purpose, provoke the camp guard, get reported and be transferred to a punishment camp. Others might have liked to cover it as far as they could, but eventually confessed when faced with incriminating letters, gifts and confessions of their lovers already extracted under duress. From the time I first read his dossier, the four words that he uttered before the guard resonated in my ears. They were: *Gefangen so oder so*', which means, 'prisoner, this way or that'. This was one of those Scottian moments of political electricity at which Gabriel blurted out the hidden transcript directly in the teeth of power (Scott 1990, p. xiii). Gabriel's utterance had a much wider and deeper resonance in the *Stalag* universe and reflected the psyche of prisoners like him who were aware of the consequences of their 'wrongdoing', but their fear of punishment was gone. It was emblematic of all hidden transcripts and desires that we can read into Gabriel C.'s love letter written in broken German. He expressed suicidal thoughts, even a death wish, if he could not unite with Frieda and their child forever. His provoked utterance and his love letter both demonstrate the passion, love and desire, combined with a sense of responsibility towards Frieda in his offer to serve at the Eastern front in exchange for her freedom. This offer was totally unacceptable to the regime whose motto was: 'once an enemy, always an enemy': intimacy with whom was absolutely forbidden.

Several dossiers of women defendants contain details of how these relationships started and blossomed. Sometimes, a POW's smile at work was reciprocated, other times a chocolate wrapping worked as a chit and was thrown at a woman in a crowded factory, quietly slipped inside her pocket or came flying through an open window inside a home to send love and propose a secret appointment. This led to Red Cross parcel-fed romances involving chocolates, cigarettes, soaps and coffee from the prisoner which were reciprocated

with home-baked food items. Laundry and mending from women would be reciprocated with plumbing, wood cutting, carpentry and fixing broken household items for gaining an entry to the beloved's home. A house key dropped in the letter box, as in Martha's case, became a tell-tale sign to her husband of a budding romance. As the parcel-fed romances became serious, the nature of the gift items changed. A wristwatch, an arm band and gold and silver items symbolizing fealty entered the list and functional items, such as a husband's clothes and accessories, were given to camouflage prisoners' identities for visiting war wives' homes, going out for long rides or in rare cases attempting an escape. Among functional gifts also figured dictionaries and lock picks, popularly known as Dietrich that facilitated communication and movement between lovers. Whatever linguistic inability they may have faced in communicating, they compensated with gifts and warm gestures to create a bond.

The last and the most remarkable emotional responses came from soldier-husbands. In my Special Court samples, in the 20 cases of war wives, 10 appealed for clemency. This roughly corresponds with Scheck's evaluation of Special Court cases from Nuremberg, Vienna and Kiel/Neumünster in which 41 out of 83 forgave their wives, 11 sought divorce and there was no information on the others (Scheck 2021, p. 265). The average in both cases turns out to be almost the same. He gives no statistics from the Berlin Brandenburg area, so it is difficult for me to make a direct comparison. However, in my regional and local trial samples, the clemency appeals were overwhelming. There were hardly any cases of divorce or revenge and all of them resulted in acquittals.

Just as Frieda's and Käte's husbands got them out of the prison as early as 1942 and 1941, respectively, another soldier-husband Bern J. hired a lawyer in February 1942 to stall the proceedings against his wife who was sentenced to 1 year penitentiary at the end of March. When the lawyer's effort proved in vain, Bern J. intervened directly in July to annul her sentence. Eventually, another communication in early November bore fruit. The Gestapo sent a communication to his wife on 24 November to warn her before her release on 17 December 1942. The appeal kept going in circles for an entire year as the Gestapo thought that she enjoyed a bad reputation in the neighborhood and did not give a favorable recommendation to the judiciary. It was the persistence of Bern J. that finally assured her release[16]. A similar example from July 1942 relates to a farmer who approached Hitler directly, while his wife sat in investigative custody, to free her in order to restore his balance and enable him to do his duty at the front. It had a great impact on Hitler, and he decided to release her. Despite that, the Special Court of Silesia sentenced her to 4 and a half years of penitentiary, a very harsh verdict. The Minister of Justice Thierack, however, wrote to the Reich's Chancellery on 3 September 1942 informing that he had suspended the sentence. (Colmorgen and Godau-Schüttke 1995, pp. 129–31)

These are just four appeals in the early years of 1941–1942, before the memo of 14th January 1943 from the Ministry of Justice advising the prosecutors to release war wives whose husbands appealed for mercy. Bern J's appeal took an entire year before he could get his wife out. These cases indicate that they were the trailblazers of a trend that gathered momentum towards the end of the war.

There was a pattern to the way in which these clemency appeals were written or articulated through other channels, such as a lawyer or kin. These were acts of doing emotions in a performative sense of involving both the consciousness and the body (Scheer 2012). They were being done with a specific purpose in mind: to decriminalize their wives' offense and obtain an acquittal for them. They were conscious that emotional appeals would be more effective when presented in a palatable manner to the law enforcers. Such presentations also involved self-censoring the provocative terms or phrases, and introducing elements such as mental and physical agony to the entire family. A typical letter or communication would begin with how the soldier-husband was very shocked and pained by his wife's infidelity and then that he confronted her. She sincerely regretted her deed, and he was convinced that she had learnt a lifetime lesson. It is only after this round of self-confessed private conflict resolution that he turned to contestations, which were

deployed in defense of her honor: virtues such as her generosity, compassion, propriety, lack of exposure to aliens, his absence for long periods which led her astray and compelled her to lean on someone else for support. Air raids and evacuations were cited as aggravating circumstances that destabilized her mental state. This would be followed by drawing attention to how she, the children and ultimately the soldier himself were suffering due to her incarceration and how his will to fight would be strengthened by her release. Having said this, these were deeply felt emotions that soldiers experienced which can be seen in the speed with which they approached the prosecutors and turned to higher authorities if they were not successful. The appeals also contained very individualistic, spontaneous and unique *eigen-sinning* elements, such as one soldier-husband offering his entire savings in return for her acquittal, blaming himself for having failed her due to his long absences and denying himself the bliss of having several children with her, while showing his willingness to sacrifice his life for the fatherland[17]. Another proposed to send his wife to the *NSV* (National Socialist People's Welfare) to volunteer her services if she was released.[18] After the initial shock, some soldier-husbands went into a self-reflective mode of not having treated their wives well, taken them for granted or simply accepting that their marriage lacked harmony but assured the prosecutor that the couple would work on bringing back the harmony and making the bond stronger after the wife's acquittal. Martha's husband, after denouncing her, tried to win her love back, but when she was arrested, he could not tolerate the separation, was overcome by guilt and committed suicide under extremely tragic circumstances. Some just refused to believe that their wives cheated on them and attributed their affairs to their loving disposition towards people around them, rather than any ill-will towards their husbands. Others alluded to the incomprehensible charms and seductive power of captives that made their wives vulnerable. Yet others believed that their wives were young and had the same sexual and emotional needs as men. An evaluation of the interiority of soldier-husbands brings them out as conflicted, self-doubting, self-reflecting and finally self-denying husbands who placed the interests of their wives above all and advocated their return with full tenacity despite being betrayed. What remains certain is that the prosecution was put in a tight spot by these clemency-seeking soldier-husbands who insisted on decriminalizing their wives' adultery and forcefully rejected the claims of the state over their bodies and freedoms. They considered the infidelity of their wives as a private matter between the couple, which they claimed they were fully capable of solving themselves. They assured the prosecution that they were responsible and capable enough to deal with a matter that affected them directly. Whatever might have gone wrong with their relationship, they wanted to sustain these long-distance marriages for the sake of their own sustenance and the love of their family, children and home that they had left behind.

Vaizey reaches the same conclusion on the basis of the *Feldpostbriefe* about the endurance of the wartime families despite the physical separation of war couples (Vaizey 2010a, 2010b). Even though the sources and contexts of the two were quite different, with *Feldpostbriefe* being the bridge that couples built to remain emotionally connected while the emotional and sexual involvement of war wives supposedly 'burnt that bridge', yet the sentiments of their husbands remained the same: to brave the odds of physical separation and sustain their conjugal life. Those who were posted in the East acted with a lingering fear of never returning, yet they claimed their thoughts were with their families, and they wanted to save them from disintegrating. The defense of the fatherland was an abstract entity for these soldier-husbands who were risking their lives on the frontline. However, when the concrete manifestation of that, namely, their home and hearth, came under attack in the name of defending the honor of the nation as well as their own, the duplicity of the state's claim and the futility of their sacrifice dawned on them. They fought back to defend the honor of their wife and their familial bond.

Indeed, the codes of honor and shame, the state-structured and executed emotions, were writ large on the trial verdicts that never failed to castigate the wives for willfully and

repeatedly violating the codes of honor by throwing themselves at enemy aliens, but they had little purchase as far as the affected people were concerned.

## 6. Conclusions

Through an in-depth analysis of the testimonies of war wives and their POW lovers, their mutual exchange of gifts as well as the clemency appeals of soldier-husbands, this paper has engaged with 'the throw' of the discursive field of criminalized intimacies. It has brought out the nuances of the *eigen-sinning* behavior of the involved persons who in their own ways defied the prescriptions and proscriptions of the state. The lovers did not perceive each other as enemy aliens. POWs often offered an alibi to shield their beloved from the consequences, and when they were pinned down due to circumstantial evidence or their beloved's confession, they would say that the two loved each other, had marriage plans and saw nothing wrong with that.

War wives felt a disconnect with the propaganda posters depicting happy mothers surrounded by several children. They were weary of constant preaching by leaders and surveillance by social workers which sought to mold them into perfect home makers, prolific mothers and sexually committed partnersof their absent soldier-husbands. They frequently aired their displeasures at the lack of romance and sex in their daily lives openly before law enforcement agencies and confessed to being attracted to the POWs who were mostly their co-workers in the factories and on the farms. Such displays of desire and agency were quite shocking for the disciplining agencies. They seemed flustered by the wartime awakening of libertarian sexuality among these 'racially healthy women with no previous criminal record'. War wives who accepted having initiated the contact were labelled the 'driving force' and punished more severely. Accusations of rape, suspected abortions and cases of delivery were investigated thoroughly, and false accusations were dismissed for lack of evidence. It was another matter that they had to release such wives on their husbands' insistence later.

Their soldier-husbands contested the scathing moral judgements in court verdicts and asserted their faith in their wives' innocence, generosity, naivety, respectability and sympathy for fellow humans. They evoked the deeply felt bond of marital love that could not be broken by fleeting dalliances between their lonely wives and seductive alien prisoners. They rallied the larger community of well-wishers to broaden their net of supporters. Community elders, siblings, parents, sympathetic doctors, lawyers and even superiors surfaced in these dossiers quite frequently to decry state measures.

These were married men whose love had transcended the intoxicating stage of falling in love that one gets to read about in the POWs' lovers' letters and testimonies. The challenge that the former faced was that of staying in love and sustaining their conjugal bond. They took it upon themselves to convince the prosecution that they were dependable, caring and responsible family men who would rebuild their home. They did so by displaying grace, forgiveness, compassion and understanding. What emerged from the heap of these clemency appeals was the haunting trope of half a man, broken by his companion's confinement. Phrases such as, 'I am just half a man without my wife'; 'my thoughts wander back to my bombarded home and neglected children who have been separated from their mother'; 'my wife and family were my life's fortune until now'; 'my wish for several children from my wife has remained unfulfilled due to my draft and now she sits behind bars totally destroyed' were commonplace in the appeals. They overcame their initial shock and grief of being wronged by their wives and soon started to worry about their troubled marriage and splintered family which they were determined to set right. They were asking the prosecution a more pertinent and fundamental question: What kind of home front (*Heimat)* were they fighting for when their own home was on fire? How could they keep their morale high with a sinking feeling that their own homes and families were faced with doom? These arguments subverted the prosecution's logic of punishing war wives for humiliating and betraying their husbands. They argued instead that they had become the ultimate sufferers of their wives' confinement. They made the defense of

the fatherland sound hollow with incarcerated wives and destroyed homes. The public sphere and state propaganda continued to project muscular, resolute, hyper-masculinized soldiers but the real soldiers could see their own nemesis when the state started devouring their own families and they needed to be at the front to face a gloomy future.

As far as the law enforcers were concerned, despite propaganda, surveillance, quick sentences and long and harsh penitentiary terms, they lost control over the 'weaker sex', and yet continued to perform their respective roles with obsessive zeal. In December 1943, an SD report bemoaned that the entire purpose of punishing war wives was being defeated by their husbands who were energetically intervening on their behalf. They were approaching the offices of Gestapo, judiciary, welfare and other state organs regularly and pleading with them not to punish their wives or to grant them clemency[19]. As late as 13 April 1944, the *SD* report was still lamenting the sinking morality of the womenfolk, especially of the war wives; their lack of racial awareness expressing frustration at the failure of their drive; and still recommending the same old formula of preaching pristine German values more effectively through films, press and radio and suspending or cutting the separation allowance of war wives[20].

In one example after another, the Hanover cases showed the prosecution's unwillingness to release the convicts, sometimes on the advice of the jail authorities, or the Gestapo. Clemency appeals were also dismissed on the ground that it was too early for penitence and such requests kept going in circles to delay the release. Invariably, the husbands then approached the Reich's Chancellery and the Ministry of Justice, and the release orders came from *RSHA* or the Ministry of Justice, Berlin. Lizzi W's husband forgave her for having sex with a French POW in the couple's house for one and a half years. Still, her case was scheduled for a hearing in the local court on 17 April 1945! Her file closed with the note that the defendant did not appear in the court[21]. This was not the Special Court, the 'panther troop' of justice: speedy, forceful and efficient in hunting down the unruly elements of society, but the local court of Berlin acting within the ambit of the normative state, while the Red Army was at the gate. This indicates that even among the law enforcers there was a mutual rivalry and contestation for power. Despite Hitler's sympathy for soldiers who wanted their wives back, and the guidelines of 14 January 1943 circulated by the Ministry of Justice, the Gestapo kept sniffing dissent, chasing offenders, gathering evidence and setting up trial dates. The judiciary kept summoning women to face changes and passing verdicts on them.

As with the Rosenstrasse protest, the cases of criminalized intimacies go on to show that in the matters of the heart, women remained resilient to threats and humiliation. When the Gestapo rounded up around 2000 Jewish men living in mixed marriages in late February 1943 and detained them at 2–4 Rosenstrasse, located in close proximity to the Gestapo headquarters in Berlin, their German wives started gathering there and did not disperse despite orders from the guards saying 'clear the streets or be shot'. Their husbands had to be released on 6 March 1943 because of 'unpleasant scenes' created by those siding with the Jews admitted Goebbels on the occasion of their release (Stoltzfus 2001). Just like these German wives living in mixed marriages, soldier-husbands too managed to get their (adulterous) wives out of the prison.

**Funding:** Revisit Fellowship of Alexander von Humboldt Stiftung/Foundation Grant Number: 1115926, APC self-funded.

**Institutional Review Board Statement:** Not applicable.

**Informed Consent Statement:** I have obtained necessary permissions by relevant archives to use the personal data on the condition of anonymizing the identities of private individuals.

**Data Availability Statement:** Not applicable.

**Conflicts of Interest:** The author declares no conflict of interest.

## Notes

1    Verordnung zur Ergänzung der Strafvorschriften zum Schutz der Wehrkraft des Deutschen Volkes vom 25. November 1939 (RGBl. I, S. 2319)

2    Verordnung über den Umgang mit Kriegsgefangenen vom 11. Mai 1940 (RGBl. I, S. 769)

3    Niedersächsisches Landesarchiv, henceforth NLA Hannover Hann 171a Hannover Acc. 107/83 Nr. 506. The names of all private individuals from all archival case files have been anonymized in keeping with theData Protection Act

4    Sächsisches Staatsarchiv, Staatsarchiv Chemnitz, 30097, Nr. 754.

5    Landesarchiv Berlin, henceforth, LAB A-Rep 341-02 Nr. 6773 (A-Rep 341-2 are cases from the Local Court proceedings).

6    LAB A-Rep 341-02 Nr. 18168.

7    Boberach, Heinz (Hrsg.). 1985. *Meldungen aus dem Reich. Die geheimen Lageberichte des Sicherheitsdienstes der SS 1938–1945.* Herrsching: Pawlak Verlag, Band 12, pp. 4530–35

8    Boberach 1985. *Meldungen aus dem Reich.* Herrsching: Pawlak Verlag. Band 15, p. 6141.

9    NLA Hannover Hann 171a Hannover Acc. 107/83 Nr. 405

10   NLA Hannover Hann 171a Hannover Acc. 107/83 Nr. 455.

11   NLA Hannover Hann 171a Hannover Acc. 107/83 Nr. 256

12   International Tracing Service Archive: ITS/ANF/KDL-Military Government Questionnaire

13   His Hanover reference is: NLA Hannover Hann 171a Hannover Acc. 107/83 Nr. 490.

14   Budesarchiv/Militärarchiv Frieburg: BArch PERS 15/10916,

15   LAB A Rep. 358-2 Nr. 6168 (A-Rep 358 cases are from the Regional Court of Berlin).

16   LAB A Rep 358-02/5485-02.

17   NLA Hannover Hann 171a Hannover Acc. 107/83 Nr. 994.

18   LAB A-Rep 341-02 Nr. 9383.

19   Boberach 1985. *Meldungen aus dem Reich.* Herrsching: Pawlak Verlag. Band 15, p. 6145.

20   Boberach 1985. *Meldungen aus dem Reich.* Herrsching: Pawlak Verlag. Band 16, pp. 6481–8.

21   LAB A Rep 341-02 Nr. 21485.

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
