# Peer review of "Criminalized Intimacies between POWs and ‘Unworthy War Wives’ and Their Soldier-Husbands’ Responses to Racial, Sexual Wartime Justice in Nazi Germany"

_genealogy, doi:10.3390/genealogy6040086_

Round 1

Reviewer 1 Report

The focus on masculinity is a promising addition to the literature. But it deserves to be placed more in the center of the stage. The image of German soldier husbands in the manuscript contradicts Nazi images of masculinity and also the notions prevalent in much of the scholarship, which is a good find. It would be important to think about the question of masculinity for the prisoners, too, however, and to discuss prisoners and German men in comparison, so as not fall into the danger of 'methodical nationalism'.

Specific comments below in the order in which the concerns arise:

- How exactly does Fraenkel's well-known approach apply to these cases? As far as French POWs were concerned, legal proceedings seem to have been followed in prosecuting the man and the woman; it was another story obviously in relations concerning eastern POWs.

- Where does the "panther toops of justice" (I assume the author means Panzer troops) reference come from?

- POWs involved with war wives were punished with penitentiary beginning sometime in 1943, but it is not true that only women associated with soldiers received a term in the penitentiary.

- The status of transformed civilian workers should be explained more clearly. Transformation did not mean naturalization.

- It would be good to explain to what extend blood tests could/count not determine paternity at this time.

- Was Willi L. really drafted in France? How so? Was he from Alsace-Lorraine?

- "Criminalized intimacies" is a helpful concept. The transnational approach is also useful and praiseworthy.

- The concept of "prison camp paradigm" denotes the erroneous view that all POWs were behind barbed wire in prison camps and therefore had no significant contact with German (and non-German) civilians. This would be important to consider in light of the postwar claims of former POWs.

- The USA could hardly be the protecting power of the British and Belgian POWs after December 1941.

- Were the Soviet or Polish POWs involved in criminalized intimacies really taken into the hands of the army? Were they not rather handed over to the Gestapo or SS?

- The case of the two Jews is puzzling. Did they not belong to a certain army?

- Were couples with deeper bonds punished more harshly? Judges in POW cases seem to have respected honest love more than flirtations and irresponsible male seduction.

- What happened after Martha's clemency plea? Was it accepted?

- P. 12: There is much information here without any reference (this is also the case elsewhere in the manuscript). Red Cross-fed romances involving chocolates, dictionaries and keys as gifts etc. Where does this information come from?

Reviewer 2 Report

A summary

This paper examines the prosecution of German soldier’s wives charged with consorting with enemy prisoners of war and the response of their husbands during the Second World War, emphasising how these cases demonstrate that POWs, German women, and their solider-husbands challenged dominant discourses of shame and honour in Nazi Germany. Beyond exploring the emotional complexity involved in POW-wife-husband relationships, the paper highlights that husbands often appealed for clemency on behalf of their wives despite the distress caused by their infidelity. While special and regional courts sought to punish women who had undermined the morale of soldiers/the racial community by consorting with the enemy, husbands argued that the imprisonment and punishment of their wives would leave them bereft of the family for which they were fighting to protect. The strength of this paper lies in revealing that while the Nazi judicial system castigated women for ‘sleeping with the enemy’, exacting punishment for fraternisation often undermined what the courts hoped to uphold: the morale of the German soldier-husband.

General concept comments

This paper presents several compelling archival tracts drawn from a larger body of research into women prosecuted for fraternising with enemy POWs/aliens in Nazi Germany. The particular strength of the paper is in the presentation of archival documentation and the description/analysis of the husbands and their clemency appeals. The paper can be improved by articulating a clearer thesis on this topic at the beginning. The reviewer found the arguments for clemency particularly interesting. While the special courts were supposedly punishing wives who had severely undermined the morale of the solider/community, solider-husbands argued that the punishment of their wives was undermining their morale. This, in the reviewer's opinion, is a very interesting dynamic/argument that should be placed at the forefront of the paper. As it stands, the argument articulated in the first half of the paper – that German women fraternised with enemy POWs despite discourses of shame and honour – is too simplistic. There is an opportunity here to present a much more nuanced and illuminating argument, one which begins to emerge in the conclusion but needs to be articulated clearly from the outset.

There are issues relating to the methodological framework, historiography, and theoretical concepts which need further consideration.

The methodology section – ‘2. Space, Context and Frame’ -- it requires attention. The paper claims to be relocating German women within the ‘working class milieu’ but the reviewer is left unsure of what this means/involves and how this has shaped the analysis of the source material. Why is this relocation necessary? Have German women been wrongly situated in a middle-class milieu? The discussion of Martin’s transnational approach is also very vague/weak and needs linking to the analysis of the source material. The link between the relocation and Martins's transnational approach needs to be better explained. In sum, The reviewer struggles to understand how this methodology has shaped the author’s approach to the source material regarding research questions, collection, and analysis. This section needs an overhaul, it is difficult to follow and understand the relevance of this section to the rest of the paper. Furthermore, references and engagement with the concepts of the working class milieu, the need to relocate women here, and transnational approaches to the study of the conflict disappear from the paper following this section.

The reviewer suggests that the source material needs to be linked to the idea of resituating German women within the ‘working class milieu’. Here, the author needs to emphasise the uniqueness of their source base compared to other authors who have examined this subject in detail, such as Scheck. It should be made clear if this is the first paper to focus on Hanover specifically and why this is a ‘working class milieu’ and not the special courts in Berlin/Vienna. The local findings could then be contrasted with findings of broader/national studies or findings in rural areas.

Rather than resituating working-class women in their milieu, it would be perhaps more effective to consider methodologies that aid the exploration of emotions/emotional histories and the difficulties/issues involved in recovering emotions for the archives. Some explicit research questions might also help the reader understand what the specific reading of the archival tracts hopes to reveal.

Some more context (only sentences, not paragraphs) regarding the system of special courts, the number of prisoners of war/foreign labourers on the home front (specifically the area the author's research has focused on, such as Hanover), and how women came into contact with aliens (certain workplaces/rural areas, etc.) would be of great benefit to readers.

The idea that women prosecuted for fraternisation with enemy aliens in Nazi Germany was the ‘birth’ of a new criminal is misleading. Fraternisation between civilian populations and enemy POWs was policed across belligerent nations, including Germany, during the First World War. There is a rich and growing literature on fraternisation during the Great War.

The utility of the concept of ‘criminalised intimacies’ and how it changes our understanding of how relations between wives and enemy POWs were conceptualised could be explained further. What does this expression mean/capture that Verbotener Umgang does not? What does the concept involve beyond describing relationships between German wives and enemy aliens? Are criminalised intimacies to be found in other contexts (beyond Nazi Germany and the Second World War)? This begins to be addressed in footnote 5 but needs to be integrated into the text if it is to be a central plank of the paper's analysis of the source material. The terminology certainly captures the relationship, but the point that this form of fraternisation (sexual) was treated with greater seriousness than others needs to be emphasised. Historians of fraternisation/POWs have argued (across different contexts) that the issue was deeply gendered and that regulations governing contract between enemy prisoners and the civil population was driven by fears of female promiscuity and ‘miscegenation’. Other authors, for example, have used phrases such as ‘illicit encounters’. In this sense, the paper is contributing to this discussion, but it could do so in a more explicit fashion.

The paper emphasises that intimate relationships between German women and enemy aliens were ‘barrier breaking’, but what barriers were broken? The paper goes on to argue that the relationships led to husbands seeking to rebuild their relationships/support their wives (attempting to reconstruct/solidify existing relationships?) Furthermore, the cases of transnational intimacies between women and allied/enemy foreigners in Germany were not the first. Relationships between ‘enemies’ have been witnessed in all wars and conflicts. Military and civil authorities during both world wars found it very difficult to police relations between women and allied/enemy aliens. It should be noted somewhere in the paper (as most historians of fraternisation do) that the archives often reveal only those who were caught or willing to reveal their relationships. Finally, the zealousness of the authorities to prosecute women charged with fraternisation compared to needs to be considered. Therefore, the extent to which the authorities were determined to prosecute women who flouted fraternisation regulations more than other criminals also needs some consideration. In the hierarchy of issues judges faced in Nazi Germany (particularly from 1944) was the infidelity of women high on the list? Was it prudent to imprison all women for these crimes or more practical to grant clemency and save policing resources?

The concept of ‘real existing soldiers’ is somewhat problematic – arguably these historical actors are being constructed through a reading of the source material. Furthermore, the paper notes that ‘deeper feelings’ of the actors ‘cannot be captured easily’. So, what is the ‘real’ and how can it be captured? While the historical actors that the author presents and constructs certainly seem to have questioned/challenged dominant discourses of honour/shame/citizenship/patriotism, these discourses were a ‘reality’ nevertheless. 

Overall, this is a fascinating and thought-provoking paper. The structure is generally sound, although the methodology section needs reworking. References to more recent work on fraternisation between POWs and women would strengthen the paper and allow it to contribute to the discussion of this topic more readily. References to archival documentation need to be tidied up to ensure readers understand the particular repository they are held at. There are some issues with phrasing/expression in the first half of the paper and the reviewer recommends a thorough edit to eliminate typos and awkward sentences. While the paper does not quite challenge previous literature related to fraternisation it does present and highlight some very compelling archival research that no doubt furthers an understanding of the emotional complexities and challenges faced by married couples separated by war and relationships between POWs and women on the home front. This paper will be of great interest to those studying fraternisation with POWs, husband/wife relations and emotional connections on the German home front, and the Nazi judicial system.

Specific comments

Line 19/20: ‘His/her interrogation of the archival sources from a people’s perspective goes into a hitherto unexplored trifle, idiosyncrasies, subjectivities, and emotions. This sentence needs rewriting or removing.  Which ‘people’ are being referred to? The reader is left unsure if this is a particular methodological approach/theory.

Line 20/22: ‘It demonstrates how the affected people caught in the barrier-breaking romance appropriated, negotiated, rejected and defied the penal code in their own ways through a display of willful conduct’. This is too vague, it needs to be rewritten and a clear thesis statement needs to be articulated.

Line 40/41: ‘by the Special Court, “the panther troops of justice” to defend the home front’. The author needs to offer the reader some more detail on the Special Courts (is the author referring to the Sondergerichte?) Some more context would help the reader understand the role of the Special Courts in the judicial system. The author goes on to note that they have utilised local court files, what is the difference between the special and local courts? When were the Special Courts first established? How did they function? What was their remit? The quote ‘the panther troops of justice’ needs to be explained – who dubbed the special courts in this manner?

Line 42: ‘a new type of female criminal’. This is not quite right as relationships between German women and foreign men (including POWs labourers) were witnessed and decried during the First World War. Lisa Todd’s work on fraternisation between German women and POWs during the First World War, in particular her chapter in Prisoners of War and Local Women in Europe and the United States, 1914-1956 edited by Matthias Reiss and Brian Feltman, needs to be considered.

Line 47:  the difference between imprisonment and penitentiary within the Nazi judicial system needs to be briefly explained.

Line 48: ‘the bare essential one’. What does this mean? Was communication between citizens and prisoners of war limited to conversations strictly about the employment/work the latter was involved in? A little more detail/explanation is needed.

Line 58: full stop needed after the footnote. Regarding footnote 3, is the following narrative regarding Frieda K. built from the Nds Hann. 171 Hann. Acc. 107/83 Nr.506 manuscript source? If so, this needs to be made clear in the whole section (lines 54 to 124 contain no references). The footnote needs reformatting to make clear which archive this is held at.

Line 65: ‘were her colleagues’, as in work colleagues? Do the proceedings reveal where Hilda P., Frieda K., and Germain worked? This is important contextual information

Line 82: ‘she go to know in Pfingston’ reads awkwardly due to the preposition ‘in’ as Pentecost is a Christian holiday, not a place. Is the reference to Pentecost/Whit (Pfingston) necessary/significant to the case?

Line 119: reference to ‘work place’ – is the nature of the employment/workplace known?

Line 145/146: It is my understanding that POWs protected by the Geneva Convention 1929 were treated more leniently and tried via court martial. The infidelity of men in wartime (in this case POWs) was more accepted than that of women. I believe that women were also expected to pay the costs of the court case whereas POWs were not. Some more specific details regarding the ‘welfare benefits’ lost and length of imprisonment would be beneficial here, or at least a reference to Scheck’s chapter in Love Between Enemies which provides an overview of the legal framework governing interactions with POWs.

Line 148/149: is this the first study of local courts? Does Scheck not cover this too? What is the significance of women being put on trial in local/regional courts as well as special courts? How does this further our understanding of the topic?

Line 156/157: Is ‘invisiblised’ a particular concept?

Line 161: why were aliens ‘not an obvious part of their memory-scape’? This needs some explanation.

Line 165/66: what is specifically meant by ‘methodological oversight’ here? Social historians favour male over female oral interviewees. Clarity needed.

Line 389: The blurring of the home front/front line and the presence of aliens/POWs on the former began with mass internment during the First World War.

Line 391/2: The assertion that proximity to aliens ‘generated a new gender dynamic’ during the Second World War is bold. Social historians have questioned the idea that ‘total war’ led to fundamental shifts in gender relations, certainly in the context of Britain. It also underplays the disruptive impact of war with family units separated. The greater freedoms and new experiences women might have enjoyed (not all women ‘enjoyed’ the experience of factory work) were tempered by the loss of their spouse to the front line. Furthermore, expectations of ‘traditional’ female behaviour were arguably heightened by the war, not loosened. The author explores how soldier husbands ‘rescued’ their wives from imprisonment by appealing to the traditional family unit and stability on the home front. In sum, this is much more complicated and nuanced than suggested.

Line 540/541: what was the image of the enemy alien? Were all nationalities of POWs viewed in the same way by the state/community? Were nationalities treated differently?  

Line 547/548: unclear what ‘resources’ and ‘mobility’ are being referred to here.

Reviewer 3 Report

I think this article has real potential, and I would very much like to see it in print. I agree with the author's criticisms of Usborne and Herbert, and am pleased that he or she has taken up some of the important claims made by Scheck regarding husbands' pleas for clemency, finding new and compelling evidence in local and regional courts. What this reveals about the Nazi state at war is gradual Kontrollverlust over the model of hegemonic masculinity established in the period of rearmament and the early phase of the war. Perhaps the author could bring out this Kontrollverlust more clearly in his or her conclusion.

My main criticism, however, and the principal reason why I am suggesting major revisions, relates to structure. This needs a major overhaul to make the piece more reader-friendly and to signpost the key claims more effectively. While I appreciate that there are some benefits to beginning with the story of Frieda K., Germain and Willi L., it may be worth moving that further down and beginning with the material in the 'literature review' section, with the author clearly positioning him or herself in relation to the existing historiography. The conclusion is also far too long. Perhaps have a section before the conclusion, with the material on how both women and men challenged the state-structured emotional responses they were expected to exhibit in the face of wartime separation and sexual/romantic deprivation. In the conclusion, which should be much briefer, I would then draw out the broader implications of this Kontrollverlust for understanding gendered forms of (in)justice in Nazi Germany. Could it be that the state's loss of control was then projected onto women 'transgressors' in court cases (and media reports) in ways which even their husbands could not bear? In short, this is a really important story that deserves to be told, but in a more structured form. In respect to content and argument, the piece has high merit and deserves to be published. But the structure needs a rethink.

On a more minor level, there are some cases of clumsy wording. 

Is 'trifle' the right word to use in the abstract? I would leave it out, as it suggests that emotions and subjectivities might normally be dismissed as trivial. The 'anxiety, fear and gloom' of the war years, referred to on p. 12, was not a trifle.

p. 3, Willi L. was stationed rather than drafted in France?

p. 8: were women who engaged in affairs or romances with aliens necessarily 'fun loving'? Is that not just another state- or media-structured emotion imposed upon them from above? If not, then where is the evidence that they were 'fun loving'? 

p. 13: Does the author mean, literally, that the trend towards clemency pleas was started with Willi L? It may be more accurate to say that he represents a poignant example of what was to become a growing trend after 1942 - as the author suggests lower down the page.

p. 13: the author says that 'criminalised intimacies' burned emotional 'bridges', but the evidence of husbands' clemency pleas surely shows that this was not always the case. In some instances, emotional bridges could be strengthened through a husband's forgiveness or even understanding and empathy.

p. 13, lower down: presumably Willy should be Willi?

Finally, one point that comes across strongly in Scheck's book is that enemy POWs, even when court martialled, were normally granted anonymity in the sense that their names were not printed in the press and their families at home did not necessarily get to hear of their 'transgressions'. Yet German women convicted in courts had their names published in the press. I was thinking about this when noticing that the French POW in the article is named as Germain (is that his surname or first name?) but the others have been granted partial anonymity by simply having the initial for surname. In the German normative state (and still today) criminal suspects have the right to such anonymity prior to conviction. So my question is, has the author him/herself chosen to just go with initials instead of full surnames, or is this how they are referred to in the court dossiers? Were husbands pleading for clemency for their wives granted anonymity (and therefore offered the protections of the normative state)?

Round 2

Reviewer 1 Report

The version is much improved, and it is impressive how quickly the author was able to revicse the manuscript. It reads much better now. It is more coherent and precise. The points raised were addressed.

A few issues remain. Some places still present information without a footnote (for example reagrding the number of French POWs in Graudenz prison or the middle paragraph on page 19). I am also still puzzled by the use of the term "panther troops" in the mss.

Author Response

Thank you very much for approving the revised submission.

  1. I have inserted Fraenkel in the appropriate place on page 10.
  2. I have also added a reference relating to Graudenz on page 16.
  3. The middle paragraph  of page 19 depicts the scenario of how the communications started between  POWs and German women  and how romance blossomed through the exchange of gifts and letters. This is a very interesting theme and as I pointed out on page 9, I would like to write another essay on it. 
  4. As for the Sondergericht being called Panther Troops of justice, it was a term that captured the spirit and instinct of the Nazi judicial system and I would like to retain the term in its present form.Thank you.

Reviewer 3 Report

I think this is a considerable improvement on the first version, especially in respect to structure. It is now much more readable, the argument is easier to follow, and the author's positioning of her or himself vis a vis the existing literature comes across with considerably greater clarity. This is a brilliant piece, and if it is published, which I very strongly recommend, then I will certainly use it in teaching.

Some minor suggestions (none of which are essential to follow)

- masculinity as a concept should perhaps appear in the abstract and as one of the keywords. This will give an even better indication to potential readers of what the piece is about

- Fraenkel has been dropped from the text, but the phrases 'prerogative state' (p. 9) and 'normative state' (p. 24) are still in use. Do they need some brief explanation?

p. 17, line 648: can he really have been a soldier in Leningrad since February 1941, i.e. before the launch of Operation Barbarossa?

Author Response

Thanks for your suggestions.

  1. I have added masculinity to the keywords.
  2. The reference to Fraenkel has been inserted in appropriate places Pages 10 and 24.
  3. Thanks for pointing out the Leningrad context. I have altered it as: soldier since February 1941 currently posted in Leningrad.                                         It has been great working with you on the submission.